# Sustainable Risk Identification Using Formal Ontologies †

Avi Shaked [1] and Oded Margalit [2,*]

1 Department of Computer Science, University of Oxford, Oxford OX1 3QD, UK
2 Department of Computer Science, Ben Gurion University of the Negev, Be'er Sheva 84105, Israel
* Correspondence: odedm@cs.bgu.ac.il
† This Paper Is an Extended Version of Our Paper Published in the Proceedings of the 17th Annual System of Systems Engineering Conference (SOSE 2022) (Rochester, NY, USA, 7–11 June 2022).

**Abstract:** The cyber threat landscape is highly dynamic, posing a significant risk to the operations of systems and organisations. An organisation should, therefore, continuously monitor for new threats and properly contextualise them to identify and manage the resulting risks. Risk identification is typically performed manually, relying on the integration of information from various systems as well as subject matter expert knowledge. This manual risk identification hinders the systematic consideration of new, emerging threats. This paper describes a novel method to promote automated cyber risk identification: OnToRisk. This artificial intelligence method integrates information from various sources using formal ontology definitions, and then relies on these definitions to robustly frame cybersecurity threats and provide risk-related insights. We describe a successful case study implementation of the method to frame the threat from a newly disclosed vulnerability and identify its induced organisational risk. The case study is representative of common and widespread real-life challenges, and, therefore, showcases the feasibility of using OnToRisk to sustainably identify new risks. Further applications may contribute to establishing OnToRisk as a comprehensive, disciplined mechanism for risk identification.

**Keywords:** formal ontology; risk identification; cybersecurity; vulnerability



## 1. Introduction

Risk identification is the process which lays the foundations for establishing the cybersecurity posture of systems, organisations and services. Risk management is a collection of "coordinated activities to direct and control an organisation with regard to risk" [1]. Risk identification provides the infrastructure for all other risk management activities [2].

A risk is a potential for something to go wrong, eventually causing harm or loss [3]. Accordingly, cyber risk is an operational risk which is associated with activities in cyberspace that may cause damage to organisational assets [4].

The goal of risk identification is to "find, recognize and describe risks that may prevent an organization achieving its objectives" [5]. Refsdal et al. identify that risk comprises three elements: asset, vulnerability and threat [3]. In agreement, Strupczewski's meta model of cyber-risk concept includes the same three elements [4]. A vulnerability merely indicates an exploitable system property; a risk is distinguished from a vulnerability by having the potential to harm or reduce the value of an asset. The identification of pertinent assets—such as sensitive information and services—and their business value is therefore an essential risk identification element [6]. Risk identification requires knowing the business environment and the organisational assets in addition to the vulnerabilities [7].

Provided risks are properly identified, they can be then analysed, evaluated for impact and, if necessary, mitigated using appropriate security controls. Otherwise, unidentified risks may go untreated, and misidentified risks may be improperly treated; potentially resulting in considerable damage once they materialise [8].

Continuous organisational changes introduce a major threat to performing risk identification [7]. The dynamics of business environments include changes to processes, products and services, as well as introduction of new information systems and related features. Irrespective of organisational changes, the cyber threat landscape is autonomously evolving. As an example, new software vulnerabilities are published on a daily basis, providing ample opportunities for attackers to exploit them [9]. Moreover, attacker capabilities—tactics, technologies and procedures (TTPs)—continue to improve [10]; sometimes to a military grade level [11]. To address the dynamics of cybersecurity, it is essential to have dynamic and adaptable cyber risk management, with risk identification outputs being revisited often to re-evaluate and establish an up-to-date organisational cybersecurity posture [6,7]. For this purpose, risk register mechanisms, such as those recommended by The European Union Agency for Cybersecurity (ENISA), contain the date of latest assessment as part of the risk register record and are expected to be properly maintained [12].

Relevant, up-to-date and timely information is crucial to robust risk identification [5]. Prevalent risk identification approaches rely on manual analysis by human experts [2]. These include brainstorming, interviews, checklists, statistics and techniques for historical data collection [3]. Risk identification also relies on integration of information from various sources [3,13]. Previous automation attempts with respect to cyber risk activities focused mostly on automated identification of threats and vulnerabilities (for example, [14,15]). Specifically, attributing the actual risk to organisational assets remains a manual analysis effort. The manual nature of risk identification approaches hinders their dynamic application in a sustainable form to meet the challenges of the evolving cybersecurity threat landscape [6].

This paper, which extends [16], proposes the use of a formal ontology to promote rigorous and continuous risk identification. A formal ontology is a well-defined, computer-based representation of concepts and their relations [17]. Formal ontology should not be confused directly with the philosophical term, which is concerned with the understanding of reality. However, formal ontology relates to the philosophical term, by capturing the ontology of a particular domain using a formal, well-structured model. We use the term "ontology" henceforth to relate to formal ontology.

Ontologies are a form of semantic technology. They provide the infrastructure for intelligent applications [18]. Ontologies belong to the content theory branch of Artificial Intelligence (AI) [19], and they are central for building intelligent computational agents [20]. Ontologies can minimise ambiguity and misunderstanding between stakeholders as well as lay the foundations for high-level reasoning and decision making [18,21]. An organisation-specific ontology can be used to facilitate interoperability between domains [22], and, even more specifically, between business and information technology concerns, with which organisational cybersecurity is typically associated [23].

Ontologies can be used to support risk management. Examples of such applications include management of human and ecological health risks [24] and safety risk management in construction [25]. Previous uses of ontologies for cybersecurity risk management did not consider the critical business impact of such risks [26–28]. An ontology-based system was demonstrated for the calculation of cybersecurity risk metrics, but it does not include inferred identification of risks and does not provide actionable risk-related information [29]. An automated security risk identification method to address engineering design issues exists, but it involves only identification of high-level consequence categories [30]. As far as we know, there is no ontology-based method to identify emerging cybersecurity risks which can be employed continuously by organisations, let alone one which allows an organisation to contextualise the risks with respect to the organisational operations.

This paper details and exemplifies a new method—OnToRisk—which uses formal ontology mechanisms to automate cybersecurity risks identification, based on integration of formal definitions and situational information from pertinent sources. OnToRisk is an AI method which employs aspects of knowledge representation to introduce robust information models; and of reasoning to provide actionable insights about situations represented by the models. The information models can include security intelligence related

concepts—namely threat, vulnerability, asset and risk—as well as any other technical and organisational concepts that are relevant to provide situational awareness.

We describe a case study of using OnToRisk to identify risks emerging from a newly published software vulnerability, in an undisclosed, international enterprise in the finance sector (henceforth, "the enterprise"). While specific, the case study is representative of a general, desirable practice in every organisation which uses software components. A software vulnerability is "an instance of a flaw, caused by a mistake in the design, development, or configuration of software, such that it can be exploited to violate some explicit or implicit security policy" [31]. While previous work by Wang and Guo used a formal ontology to analyse vulnerabilities from the technical perspective of vulnerability management [21]; our case study uses a formal ontology to capture concepts and relations to analyse cybersecurity vulnerabilities from the organisational operations risk perspective.

The paper continues as follows. Section 2 presents the new, ontology-based risk identification method OnToRisk and overviews the vulnerability-induced risks identification case study. Section 3 details the case study results of using OnToRisk for vulnerability-induced risks identification. Section 4 reflects on the new risk identification method and the case study, as well as discusses further uses, benefits and research potential of the ontology-based method.

## 2. Materials and Methods

OnToRisk uses formal ontology mechanisms for rigorous, information-based and definitions-based risks identification. The OnToRisk method includes the following activities:

1. formally define concepts associated with a specific risk type, as well as their relations, by authoring an ontology;
2. formally define the risk type in the ontology, using the predefined concepts and relations. This definition of a risk type aims to promote the automatic identification of its instantiations;
3. capture the organisational situation by instantiating the existing ontology definitions. This is achieved by incorporating "individual" definitions into the ontology;
4. apply automated, ontology-based reasoners to the ontology to derive new, inferred insights about the situation.

Activity #3 is meant to be automated as much as possible, e.g., by importing—while translating—existing organisational information from information systems into the formal ontology. Activity #4 is the activity in which new risk-related insights should emerge, automatically, based on the integration of explicit definitions and explicit situations. Ideally, these activities should be performed continuously, reflecting an up-to-date organisational security posture.

We validate OnToRisk using a case study methodology. The selection in a single-case study approach is aligned with the rationale identified by Yin; that the case study is a representative, typical case [32]. The OnToRisk method is applied in a case study of an enterprise seeking to identify risks emerging from the disclosure of a new vulnerability, which is found in a prevalent software component. The widely representative and applicable case study was inspired by real events, following the late 2021 disclosure of a vulnerability in Log4j [33,34].

Risk management is considered a business-related activity in an enterprise. Accordingly, the enterprise established and maintains a system of policies, as well as a hierarchical framework for communicating and assessing operational risks, with cybersecurity risks being included as part of the overall risk management organisational system. The risk-related concepts were identified based on careful reading of official documents and directives, analysis of some of the enterprise's information systems, and on conversations with domain experts. The latter included risk managers and an incident response leader.

First, as the OnToRisk method outlines, relevant concepts and their relations were defined as a formal ontology. Protégé was the tool used for authoring the ontology [35]. The ontology itself is in the standard Web Ontology Language (OWL) format. Relevant concepts are depicted in OWL using "classes"; and relations between concepts are formally

expressed in OWL using "object properties". In defining object properties, the source node class is referred to as "Domain," and the target node class is referred to as "Range".

A relevant risk definition was then added to the ontology, using some of the predefined classes and object properties. Next, a situation was captured. The situation was designed using natural language, and then translated into the ontology, as an instantiation of the formalised classes and object properties. Finally, a reasoner (HermiT within Protégé) is used to reason about the situation, i.e., process the explicit situation definitions and present inferred information based on these. The inference was verified to yield the results that are expected based on manual analysis of the situation.

The work, including the ontology and the resulting insight with respect to the enterprise's operations and infrastructure, was presented to domain experts as well as high-level management for both obtaining feedback and promoting the organisational risk management practices.

## 3. Results

We now describe the results of applying OnToRisk to the case study (of identifying risks to the enterprise as they emerge from the disclosure of a new vulnerability in a software component). Appendix A provides the full definitions, described in Sections 3.1–3.3, in the form of a formal ontology. Appendix B provides the inferred assertions, described in Section 3.4, in the form of a formal ontology.

### 3.1. Concepts and Relations (Meta Levels Definitions)

Figure 1 shows the concepts and relations, representing the result of performing activity #1 of OnToRisk in the case study. Concepts (classes) appear as graph nodes and relations (object properties) appear as edges between nodes. The concepts are:

1. Application, representing a software application by the enterprise;
2. Component, representing any software component;
3. Business Function, representing any function that relates to the enterprise's business operation;
4. Sensitive Information, representing any sensitive information item owned by the enterprise;
5. Vulnerability, representing any vulnerability of software components;
6. Risk, representing the enterprise's risk definitions;
7. Cybersecurity Risk, representing a specific subclass of risk definitions relating to cybersecurity issues;
8. Vulnerability-Induced Risk, representing any risk to the business emerging from the existence of a vulnerability. Being a risk definition relating to a cybersecurity issue, it is a specific subclass of Cybersecurity Risk.

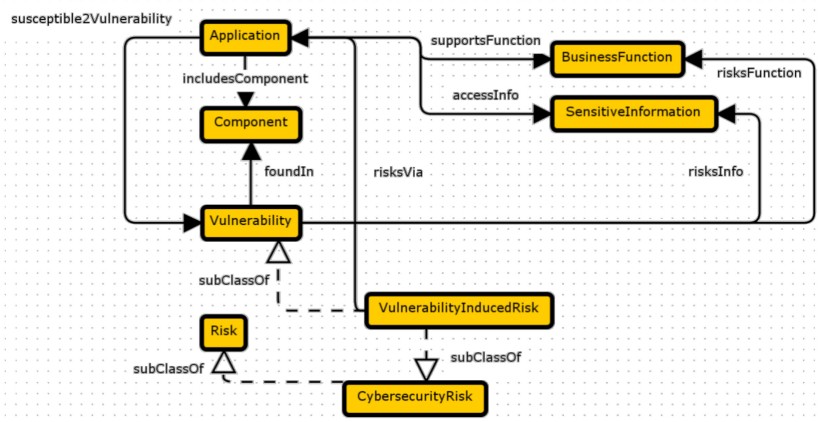

**Figure 1.** The vulnerability-induced risk case represented as a formal graph of meta-level concepts and relations. This graph is generated by applying the CoModIDE plugin for Protégé [36] to the formal ontology. The graph shows: the classes as nodes; the object properties between classes (from domain to range) as solid, annotated arrows; and subtype (subclass) relations as dashed arrows.

The hierarchical structure of the risk concepts (concepts #6, #7 and #8 above) reflects the hierarchical risk definition architecture which is practiced within the enterprise; with Risk being a layer-1 risk definition, Cybersecurity risk being a layer-2 risk definition, and Vulnerability-induced Risk being introduced as a layer-3 risk definition. This conforms with the prominent business risk typology used in the financial sector with which the enterprise is associated [4].

Figure 1 also shows the object properties—expressing relations between concepts—as graph edges in solid line between the class nodes. The object properties are:

1. accessInfo—represents an ability of an application (Domain) to access a sensitive information item (Range);
2. supportsFunction—represents that an application (Domain) supports a business function (Range);
3. includesComponent—represents a software application composition, linking the application (Domain) with its components (Range);
4. foundIn—represents a vulnerability (Domain) found in a software component (Range);
5. susceptible2Vulnerability—marks an application (Domain) as being susceptible to a vulnerability (Range) due to one of its software components. This object property is formally defined as a composite property using other object properties:

$$\text{susceptible2Vulnerability} \equiv \text{inverse(foundIn)} \circ \text{includesComponent} \tag{1}$$

6. risksInfo—indicates that a vulnerability (Domain) may risk sensitive information (Range). This object property is formally defined as a composite property using other object properties:

$$\text{risksInfo} \equiv \text{accessInfo} \circ \text{inverse(includesComponent)} \circ \text{foundIn} \tag{2}$$

7. risksFunction—indicates that a vulnerability (Domain) may risk a business function (Range). This object property is formally defined as a composite property using other object properties:

$$\text{risksFunction} \equiv \text{supportsFunction} \circ \text{inverse(includesComponent)} \circ \text{foundIn} \tag{3}$$

8. risksVia—identifies the application (Range) through which a specific Vulnerability-Induced Risk (Domain) can be realised. This object property is formally defined as a composite property using other object properties:

$$\begin{aligned} \text{risksVia} \equiv\ &\text{inverse(accessInfo)} \circ \text{risksInfo} \,| \\ &\text{inverse(supportsFunction)} \circ \text{risksFunction} \end{aligned} \tag{4}$$

Reflecting on the derived ontology, we note that it is realistic and practical to acquire relevant information, which can be used for instantiating a situation using the ontology meta-level definitions. The enterprise operates an information system which records all the enterprise applications, along with attributes. Some of these attributes are the category of information that can be accessed by the application; and the application's business criticality score, which is established based on supported business functions. Extracting software components used by an application—a Software Bill Of Material (SBOM)—is a feature provided by various software composition analysis tools (by analysing either the source code or the final software artifacts). Information about software components vulnerabilities is found online in vulnerability repositories, such as [37].

### 3.2. Risk Definition

Following the activity #2 guideline of the OnToRisk method, a Vulnerability-Induced Risk concept is formally defined using the established concepts and relations:

$$\text{vulnerabilityInducedRisk} \equiv \text{vulnerability and}$$
$$((\text{risksFunction some BusinessFunction}) \text{ or} \qquad (5)$$
$$(\text{risksInfo some SensitiveInformation}))$$

This formally defines the specific risk as a vulnerability which risks either a business function and/or sensitive information. Ideally, this definition could instantiate new information elements (of the VulnerabilityInducedRisk type). However, due to limitation in both the OWL ontology standard and the Protégé ontology authoring tool, instantiation of new elements is not possible by inference, and instead this tags a Vulnerability typed individual element as a VulnerabilityInducedRisk. Accordingly, VulnerabilityInducedRisk is also considered as a subclass of Vulnerability (in addition to being a subclass of Cybersecurity Risk); this is shown in Figure 1. This is merely a technical adaptation, which has no effect on the results as it can be easily interpreted to the ideal case, and we discuss this shortly. The formal definition of the set of risks (*R*) in this implementation is:

$$R \equiv \{v \in V \mid (\exists x \in BF \ \& \ (v, x) \in RF) \ or \ (\exists y \in SI \ \& \ (v, y) \in RI)\} \qquad (6)$$

with:
*V*—the set of Vulnerability class (i.e., concept) instantiations
*BF*—the set of Business Function class instantiations
*RF*—the set of risksFunction object properties instantiations
*SI*—the set of Sensitive Information class instantiations
*RI*—the set of risksInfo object properties instantiations
i.e., the set of risks is a subset of all vulnerabilities that have either a risksFunction object property (stating the vulnerability risks an existing business function) or a riskInfo object property (stating the vulnerability risks existing sensitive information).

The formal definition itself is more than a technical definition. This is the first concrete layer-3 risk definition, which extends the existing conceptual and abstract layer-2 enterprise risk definition (Cybersecurity risk). This fairly simple, formal ontology-based definition of a "vulnerability-induced risk" rigorously expresses a concrete type of risk. This specific risk type is of high importance to enterprise stakeholders, including its high-level management, and had not been declared until our OnToRisk implementation named it explicitly.

### 3.3. Situation

The case study situation details a risk assessment scenario which considers a newly disclosed vulnerability. It is based on real-life situations—specifically, the discovery and public disclosure of the vulnerability known as "Log4shell" [33,34]. The case study is designed as an alternative, what-if scenario of detecting risks associated with the vulnerability using OnToRisk.

According to OnToRisk activity #3, the situation is captured as a collection of instantiations of the ontological concepts and relations (derived in activity #1 and reported in Section 3.1).

The baseline situation captures the organisational situation with respect to its operational applications and their business context. Four applications exist:

1.  App1, which does not include Log4j as one of its software components;
2.  App2, which includes Log4j as one of its software components;
3.  App3, which includes Log4j as one of its software components and has access to the sensitive information item named ClientIDsList;
4.  App4, which includes Log4j as one of its software components and supports the business function named OpenAccount.

Now, consider the publication of a new Common Vulnerabilities and Exposures (CVE) record, related to the Log4j component. This results in a new situation, captured in formal ontology form by adding the newly disclosed vulnerability into our ontology, as an instantiation of the "Vulnerability" concept. We name this entity "Log4shell." Additionally, the vulnerability is associated with the affected software component—Log4j—by adding a "foundIn" object property from the Log4shell individual to the Log4j individual.

### 3.4. Ontology-Based Inferrence

According to OnToRisk activity #4, we use the ontology-based reasoner to make inferences about the developing situation, and, ultimately, identify the emerging risks. The resulting inferred assertions that extend the explicitly declared assertions appear in Appendix B.

The reasoner provides the following new inferences:

1. The "susceptible2Vulnerability" object property is attributed to App2, App3 and App4. This suggests that each of these applications is susceptible to the vulnerability.
2. The Log4shell vulnerability is categorized—automatically—as a VulnerabilityInducedRisk. This indicates that this specific vulnerability introduces new risk/s to the enterprise, as Figure 2 shows. This is the automatic identification of new risks.
3. The object property "risksInfo ClientIDsList" emerges with respect to the Log4shell vulnerability (Figure 2). This suggests that ClientIDsList, which is one of the enterprise's sensitive information items, is at risk.
4. The object property "risksFunction OpenAccount" emerges with respect to the Log4shell vulnerability (Figure 2). This suggests that OpenAccount—one of the enterprise's business functions—is at risk.
5. Two new risksVia object property assertions emerge, with respect to the Log4shell vulnerability (Figure 2). Each of these suggests a possible attack surface through which the risk can realise. In the specific case, App3 is the attack surface for the risk on ClientIDsList and App4 is the attack surface for the risk on the OpenAccount. While this is not captured explicitly in the inferred assertions, the reasoner explanation mechanism provides this traceability, as Figure 3 shows.

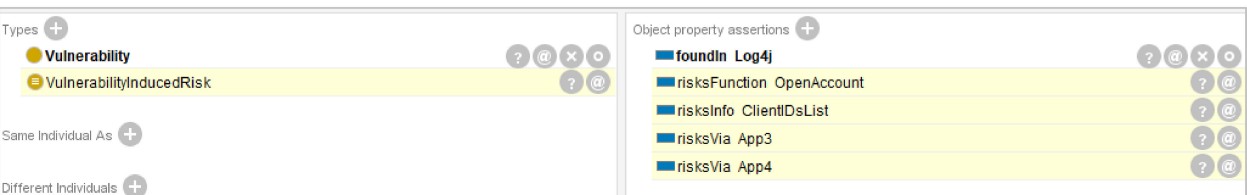

**Figure 2.** The Log4shell ontology-based assertions in Protege. Manually stated (explicit) assertions appear in bold font, while automatically inferred assertions appear in regular font.

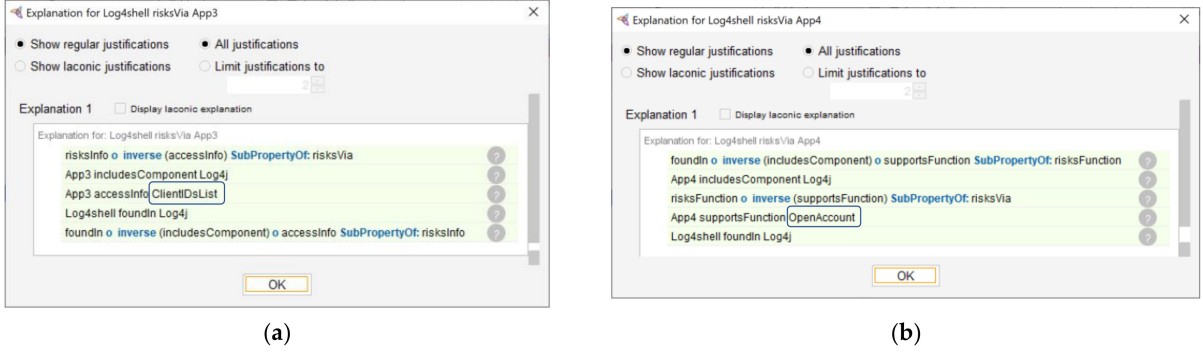

(**a**)  (**b**)

**Figure 3.** The reasoner explanation for asserting the risksVia properties. (**a**) for App3, as a result of the risksInfo with respect to ClientIDsList; (**b**) for App4, as a result of the risksFunction with respect to OpenAccount.

The automatically derived inferences are aligned with a manual analysis of the situation. While the manual analysis can be considered straightforward, performing such an analysis is time consuming, and this is exactly the effort that OnToRisk is designed to make redundant. The vulnerability in App2 does not present a new risk to the enterprise, from an operational perspective. Still, using the "susceptible2Vulnerability" property which now characterises the application, the potential of App2 being affected by the vulnerability can be communicated with the App2 application owner. The application owner can then choose whether to further analyse the vulnerability impact on the application and/or solve any vulnerability-related issues in a future version. The vulnerabilities in App3 and in App4, however, should be of interest to the enterprise management, as they introduce business risks. Continuously applying the reasoner to the enterprise situation allows pertinent managers to be notified immediately of such risks as they emerge; and the enterprise management can then promptly act to solve them, by identifying and empowering the appropriate personnel—such as application owners, risk managers, security officers and information officers—to do so.

## 4. Discussion

The dynamic cybersecurity threat landscape requires risk identification to be performed continuously to achieve up-to-date situational awareness. This paper proposes a new, formal ontology-based method—OnToRisk—for promoting automated risk identification. The method relies on the use of AI—through its formal ontology branch—for information-based, systematic and continuous risk identification. The method employs formal ontology definitions of domain concepts and relations, as well of the associated risk, to analyse organisational situations and automatically provide actionable insights.

The OnToRisk method was successfully applied to identify risks emerging from a vulnerability disclosure, which is a widely applicable challenge in enterprises. As a given enterprise situation has changed to reflect existence of a new vulnerability, a reasoning mechanism—applied to the situation—automatically yielded a list of potentially affected applications as well as of the potential business impact. In practice, typical software applications may include hundreds of re-used lower-level components, which may lead to a significant effort in their manual analysis. The automated approach of OnToRisk decouples the risk identification effort from the quantity of software components. Moreover, new risks are identified, along with their potential business impact and the respective attack surface. A reasoning mechanism can act continuously on the information. These provide a strong basis for sustainable risk management, which is essential to creating a valid cybersecurity situational awareness.

Our method provides a step forward with respect to a previously identified need for a conceptual framework to drive the rapid and automated integration of Cyber Threats Intelligence (CTI) [10]. Specifically, our method conforms with the requirement that both internal and external information be factored into the automated integration process; and it provides a rigorous infrastructure for such integration. The case study demonstrates the integration of internal, enterprise-owned information—about applications composition as well as about their business context—with external vulnerability information. Currently, some of the data was integrated manually, by importing data—exported from various information systems—into the ontology. In the case study, information about enterprise applications was adopted from the enterprise's information system which is used to catalogue applications and their metadata. A likely technical future effort is to develop mechanisms to automate the integration of data into the ontology, using both internal data sources (such as application inventory information systems) and external data sources (such as CVE repositories).

Furthermore, with OnToRisk being a technology-agnostic and vendor-neutral method, the formal representation of a domain of interest may lead to identification of gaps in information, which in turn may justify the introduction of new technology and/or tools into the enterprise. Specifically, the case study's formal ontology relies on associating each application with its SBOM. However, at the time of performing the case study, the

enterprise has only employed SBOM tools to ingest open-source software packages and did not apply the relevant technology to produce the SBOM of its own applications. Our case study highlights the need to incorporate the technology and tools to extract SBOM from the enterprise applications that are in production in order to support risk assessment with respect to vulnerability-induced risks.

In the technical implementation of the case study, vulnerability-induced risks are represented by "tagging" vulnerabilities as vulnerability-induced risk, i.e., the risks are a subset of the vulnerabilities (as captured formally in Equation (6)). This is due to limitations in the OWL standard and the standard Protégé implementation that prevents inferring the existence of new individuals. We chose to adhere to the standard implementation to demonstrate the feasibility and practicality of OnToRisk. Ideally, however, the risk identification implementation can be easily improved when developing an ontology-based application or information system by using a proprietary mechanism to yield new individuals. Such individuals can be derived formally as the tuple (vulnerability, impacted element, attack surface), i.e.:

$$
\begin{aligned}
(v,i,a) \equiv \{(v,i,a)| \\
(\ni v \in V \ \& \ \ni i \in BF \ \& \ \ni a \in A \ \& \ (a,i) \in SF \ \& \ (v,i) \in \ RF \ \& \ (v,\ a) \in RV)\ or \\
(\ni v \in V \ \& \ \ni i \in SI \ \& \ \ni a \in A \ \& \ (a,i) \in AI \ \& \ (v,i) \in \ RI \ \& \ (v,\ a) \in RV)\}
\end{aligned} \tag{7}
$$

with:

  $V$—the set of Vulnerability class (i.e., concept) instantiations
  $BF$—the set of Business Function class instantiations
  $A$—the set of Application class instantiations
  $SF$—the set of supportsFunction object properties instantiations
  $RF$—the set of risksFunction object properties instantiations
  $SI$—the set of Sensitive Information class instantiations
  $AI$—the set of accessInfo object properties instantiations
  $RI$—the set of risksInfo object properties instantiations
  $RV$—the set of risksVia object properties instantiations

OnToRisk currently provides the identification of potential risks. Identified risks should be further analysed. In the case study implementation, for example, an application marked as susceptible to a vulnerability due to the identity of one of its components may not present an actual risk, e.g., in a case where an application uses the component in a version which is not susceptible to the vulnerability or if the context of use or security controls prevent the exploitation of the disclosed vulnerability. Future research can establish the use of other ontology elements, such as data properties—in addition to classes and object properties—for improving the risk identification and its automation. Expanding the ontology with additional elements may also contribute to the prioritisation of risks (e.g., by introducing impact levels) and to the inclusion of additional CTI information.

With OnToRisk currently validated for the specific case study of a vulnerability-induced risk, additional research can utilise the method to identify other types of cybersecurity risks, such as those emerging from a compromised supply chain or from existence of Common Weaknesses Enumeration (CWE) in applications and application development.

Whereas a previous method by Eckhart et al. employs automated risk identification for improving engineering artifacts [30], OnToRisk provides automated risk identification for better organisational situational awareness. OnToRisk provides a more concrete view of business consequences, compared with the high-level consequence categories of the engineering-focused method proposed by Eckhart et al. OnToRisk relies on continuous integration of information within operational context, as opposed to initiated engineering design verification, which is the domain of the method by Eckhart et al. Both methods share a formal ontology approach as well as the goal of relieving personnel from tedious risk identification so that it can concentrate on other aspects of risk management. Therefore, future research may seek to integrate the two methods and deliver an ontology-based risk identification method for the full system lifecycle.

## 5. Conclusions

In this paper we describe a new method—OnToRisk—which promotes the automatic identification of risks. The method is validated using a widely applicable, realistic and representative case study implementation of identifying risks emerging from software vulnerabilities.

Future research may demonstrate the use of the proposed method to support the automated identification of risks of additional types. Furthermore, elaborating the ontology definitions and the ontology-based reasoning can improve the output of the method, providing a more accurate and prioritised risk identification.

**Author Contributions:** Conceptualization, A.S.; methodology, A.S.; software, A.S.; validation, A.S.; resources, O.M.; writing—original draft preparation, A.S.; writing—review and editing, O.M.; visualization, A.S.; project administration, O.M. All authors have read and agreed to the published version of the manuscript.

**Funding:** This research received no external funding.

**Institutional Review Board Statement:** Not Applicable.

**Informed Consent Statement:** Not Applicable.

**Data Availability Statement:** Not applicable.

**Conflicts of Interest:** The authors declare no conflict of interest.

## Appendix A. The Case Study Formal Ontology (OWL Format)

This appendix provides the full ontology of the reported case study. The results are fully reproducible by copying the ontology into a text file and opening it with the Protégé ontology authoring tool.

```xml
<?xml version="1.0"?>
<Ontology xmlns="http://www.w3.org/2002/07/owl#">
  <Prefix name="owl" IRI="http://www.w3.org/2002/07/owl#"/>
  <Declaration>
    <Class IRI="owlapi:ontology578765402008551#Risk"/>
  </Declaration>
  <Declaration>
    <Class IRI="owlapi:ontology578765402008553#CybersecurityRisk"/>
  </Declaration>
  <Declaration>
    <Class IRI="owlapi:ontology578765402008555#VulnerabilityInducedRisk"/>
  </Declaration>
  <Declaration>
    <Class IRI="owlapi:ontology578765402008557#Application"/>
  </Declaration>
  <Declaration>
    <Class IRI="owlapi:ontology578765402008559#Component"/>
  </Declaration>
  <Declaration>
    <Class IRI="owlapi:ontology578765402008561#BusinessFunction"/>
  </Declaration>
  <Declaration>
    <Class IRI="owlapi:ontology578765402008563#SensitiveInformation"/>
  </Declaration>
  <Declaration>
    <Class IRI="owlapi:ontology578765402008565#Vulnerability"/>
  </Declaration>
```

```
<Declaration>
   <ObjectProperty IRI="owlapi:ontology578765402008567#includesComponent"/>
</Declaration>
<Declaration>
   <ObjectProperty IRI="owlapi:ontology578765402008569#foundIn"/>
</Declaration>
<Declaration>
   <ObjectProperty IRI="owlapi:ontology578765402008571#risksFunction"/>
</Declaration>
<Declaration>
   <ObjectProperty IRI="owlapi:ontology578765402008573#risksInfo"/>
</Declaration>
<Declaration>
   <ObjectProperty IRI="owlapi:ontology578765402008577#supportsFunction"/>
</Declaration>
<Declaration>
   <ObjectProperty IRI="owlapi:ontology578765402008581#accessInfo"/>
</Declaration>
<Declaration>
   <ObjectProperty IRI="owlapi:ontology578765402008588#risksVia"/>
</Declaration>
<Declaration>
   <ObjectProperty IRI="owlapi:ontology578765402008590#susceptible2Vulnerability"/>
</Declaration>
<Declaration>
   <NamedIndividual IRI="http://www.co-ode.org/ontologies/ont.owl#App1"/>
</Declaration>
<Declaration>
   <NamedIndividual IRI="http://www.co-ode.org/ontologies/ont.owl#App2"/>
</Declaration>
<Declaration>
   <NamedIndividual IRI="http://www.co-ode.org/ontologies/ont.owl#App3"/>
</Declaration>
<Declaration>
   <NamedIndividual IRI="http://www.co-ode.org/ontologies/ont.owl#App4"/>
</Declaration>
<Declaration>
   <NamedIndividual IRI="http://www.co-ode.org/ontologies/ont.owl#ClientIDsList"/>
</Declaration>
<Declaration>
   <NamedIndividual IRI="http://www.co-ode.org/ontologies/ont.owl#Log4j"/>
</Declaration>
<Declaration>
   <NamedIndividual IRI="http://www.co-ode.org/ontologies/ont.owl#Log4shell"/>
</Declaration>
<Declaration>
   <NamedIndividual IRI="http://www.co-ode.org/ontologies/ont.owl#OpenAccount"/>
</Declaration>
<EquivalentClasses>
   <Class IRI="owlapi:ontology578765402008555#VulnerabilityInducedRisk"/>
   <ObjectIntersectionOf>
      <Class IRI="owlapi:ontology578765402008565#Vulnerability"/>
      <ObjectUnionOf>
         <ObjectSomeValuesFrom>
```

```xml
            <ObjectProperty IRI="owlapi:ontology578765402008571#risksFunction"/>
            <Class IRI="owlapi:ontology578765402008561#BusinessFunction"/>
          </ObjectSomeValuesFrom>
          <ObjectSomeValuesFrom>
            <ObjectProperty IRI="owlapi:ontology578765402008573#risksInfo"/>
            <Class IRI="owlapi:ontology578765402008563#SensitiveInformation"/>
          </ObjectSomeValuesFrom>
        </ObjectUnionOf>
      </ObjectIntersectionOf>
    </EquivalentClasses>
    <SubClassOf>
      <Class IRI="owlapi:ontology578765402008553#CybersecurityRisk"/>
      <Class IRI="owlapi:ontology578765402008551#Risk"/>
    </SubClassOf>
    <SubClassOf>
      <Class IRI="owlapi:ontology578765402008555#VulnerabilityInducedRisk"/>
      <Class IRI="owlapi:ontology578765402008553#CybersecurityRisk"/>
    </SubClassOf>
    <SubClassOf>
      <Class IRI="owlapi:ontology578765402008555#VulnerabilityInducedRisk"/>
      <Class IRI="owlapi:ontology578765402008565#Vulnerability"/>
    </SubClassOf>
    <ClassAssertion>
      <Class IRI="owlapi:ontology578765402008557#Application"/>
      <NamedIndividual IRI="http://www.co-ode.org/ontologies/ont.owl#App1"/>
    </ClassAssertion>
    <ClassAssertion>
      <Class IRI="owlapi:ontology578765402008557#Application"/>
      <NamedIndividual IRI="http://www.co-ode.org/ontologies/ont.owl#App2"/>
    </ClassAssertion>
    <ClassAssertion>
      <Class IRI="owlapi:ontology578765402008557#Application"/>
      <NamedIndividual IRI="http://www.co-ode.org/ontologies/ont.owl#App3"/>
    </ClassAssertion>
    <ClassAssertion>
      <Class IRI="owlapi:ontology578765402008557#Application"/>
      <NamedIndividual IRI="http://www.co-ode.org/ontologies/ont.owl#App4"/>
    </ClassAssertion>
    <ClassAssertion>
      <Class IRI="owlapi:ontology578765402008563#SensitiveInformation"/>
      <NamedIndividual IRI="http://www.co-ode.org/ontologies/ont.owl#ClientIDsList"/>
    </ClassAssertion>
    <ClassAssertion>
      <Class IRI="owlapi:ontology578765402008559#Component"/>
      <NamedIndividual IRI="http://www.co-ode.org/ontologies/ont.owl#Log4j"/>
    </ClassAssertion>
    <ClassAssertion>
      <Class IRI="owlapi:ontology578765402008565#Vulnerability"/>
      <NamedIndividual IRI="http://www.co-ode.org/ontologies/ont.owl#Log4shell"/>
    </ClassAssertion>
    <ClassAssertion>
      <Class IRI="owlapi:ontology578765402008561#BusinessFunction"/>
      <NamedIndividual IRI="http://www.co-ode.org/ontologies/ont.owl#OpenAccount"/>
    </ClassAssertion>
```

```xml
<ObjectPropertyAssertion>
  <ObjectProperty IRI="owlapi:ontology578765402008567#includesComponent"/>
  <NamedIndividual IRI="http://www.co-ode.org/ontologies/ont.owl#App2"/>
  <NamedIndividual IRI="http://www.co-ode.org/ontologies/ont.owl#Log4j"/>
</ObjectPropertyAssertion>
<ObjectPropertyAssertion>
  <ObjectProperty IRI="owlapi:ontology578765402008567#includesComponent"/>
  <NamedIndividual IRI="http://www.co-ode.org/ontologies/ont.owl#App3"/>
  <NamedIndividual IRI="http://www.co-ode.org/ontologies/ont.owl#Log4j"/>
</ObjectPropertyAssertion>
<ObjectPropertyAssertion>
  <ObjectProperty IRI="owlapi:ontology578765402008581#accessInfo"/>
  <NamedIndividual IRI="http://www.co-ode.org/ontologies/ont.owl#App3"/>
  <NamedIndividual IRI="http://www.co-ode.org/ontologies/ont.owl#ClientIDsList"/>
</ObjectPropertyAssertion>
<ObjectPropertyAssertion>
  <ObjectProperty IRI="owlapi:ontology578765402008567#includesComponent"/>
  <NamedIndividual IRI="http://www.co-ode.org/ontologies/ont.owl#App4"/>
  <NamedIndividual IRI="http://www.co-ode.org/ontologies/ont.owl#Log4j"/>
</ObjectPropertyAssertion>
<ObjectPropertyAssertion>
  <ObjectProperty IRI="owlapi:ontology578765402008577#supportsFunction"/>
  <NamedIndividual IRI="http://www.co-ode.org/ontologies/ont.owl#App4"/>
  <NamedIndividual IRI="http://www.co-ode.org/ontologies/ont.owl#OpenAccount"/>
</ObjectPropertyAssertion>
<ObjectPropertyAssertion>
  <ObjectProperty IRI="owlapi:ontology578765402008569#foundIn"/>
  <NamedIndividual IRI="http://www.co-ode.org/ontologies/ont.owl#Log4shell"/>
  <NamedIndividual IRI="http://www.co-ode.org/ontologies/ont.owl#Log4j"/>
</ObjectPropertyAssertion>
<ObjectPropertyDomain>
  <ObjectProperty IRI="owlapi:ontology578765402008567#includesComponent"/>
  <Class IRI="owlapi:ontology578765402008557#Application"/>
</ObjectPropertyDomain>
<ObjectPropertyDomain>
  <ObjectProperty IRI="owlapi:ontology578765402008569#foundIn"/>
  <Class IRI="owlapi:ontology578765402008565#Vulnerability"/>
</ObjectPropertyDomain>
<ObjectPropertyDomain>
  <ObjectProperty IRI="owlapi:ontology578765402008571#risksFunction"/>
  <Class IRI="owlapi:ontology578765402008565#Vulnerability"/>
</ObjectPropertyDomain>
<ObjectPropertyDomain>
  <ObjectProperty IRI="owlapi:ontology578765402008573#risksInfo"/>
  <Class IRI="owlapi:ontology578765402008565#Vulnerability"/>
</ObjectPropertyDomain>
<ObjectPropertyDomain>
  <ObjectProperty IRI="owlapi:ontology578765402008577#supportsFunction"/>
  <Class IRI="owlapi:ontology578765402008557#Application"/>
</ObjectPropertyDomain>
<ObjectPropertyDomain>
  <ObjectProperty IRI="owlapi:ontology578765402008581#accessInfo"/>
  <Class IRI="owlapi:ontology578765402008557#Application"/>
</ObjectPropertyDomain>
```

```
<ObjectPropertyDomain>
  <ObjectProperty IRI="owlapi:ontology578765402008588#risksVia"/>
  <Class IRI="owlapi:ontology578765402008555#VulnerabilityInducedRisk"/>
</ObjectPropertyDomain>
<ObjectPropertyDomain>
  <ObjectProperty IRI="owlapi:ontology578765402008590#susceptible2Vulnerability"/>
  <Class IRI="owlapi:ontology578765402008557#Application"/>
</ObjectPropertyDomain>
<ObjectPropertyRange>
  <ObjectProperty IRI="owlapi:ontology578765402008567#includesComponent"/>
  <Class IRI="owlapi:ontology578765402008559#Component"/>
</ObjectPropertyRange>
<ObjectPropertyRange>
  <ObjectProperty IRI="owlapi:ontology578765402008569#foundIn"/>
  <Class IRI="owlapi:ontology578765402008559#Component"/>
</ObjectPropertyRange>
<ObjectPropertyRange>
  <ObjectProperty IRI="owlapi:ontology578765402008571#risksFunction"/>
  <Class IRI="owlapi:ontology578765402008561#BusinessFunction"/>
</ObjectPropertyRange>
<ObjectPropertyRange>
  <ObjectProperty IRI="owlapi:ontology578765402008573#risksInfo"/>
  <Class IRI="owlapi:ontology578765402008563#SensitiveInformation"/>
</ObjectPropertyRange>
<ObjectPropertyRange>
  <ObjectProperty IRI="owlapi:ontology578765402008577#supportsFunction"/>
  <Class IRI="owlapi:ontology578765402008561#BusinessFunction"/>
</ObjectPropertyRange>
<ObjectPropertyRange>
  <ObjectProperty IRI="owlapi:ontology578765402008581#accessInfo"/>
  <Class IRI="owlapi:ontology578765402008563#SensitiveInformation"/>
</ObjectPropertyRange>
<ObjectPropertyRange>
  <ObjectProperty IRI="owlapi:ontology578765402008588#risksVia"/>
  <Class IRI="owlapi:ontology578765402008557#Application"/>
</ObjectPropertyRange>
<ObjectPropertyRange>
  <ObjectProperty IRI="owlapi:ontology578765402008590#susceptible2Vulnerability"/>
  <Class IRI="owlapi:ontology578765402008565#Vulnerability"/>
</ObjectPropertyRange>
<SubObjectPropertyOf>
  <ObjectPropertyChain>
    <ObjectProperty IRI="owlapi:ontology578765402008567#includesComponent"/>
    <ObjectInverseOf>
      <ObjectProperty IRI="owlapi:ontology578765402008569#foundIn"/>
    </ObjectInverseOf>
  </ObjectPropertyChain>
  <ObjectProperty IRI="owlapi:ontology578765402008590#susceptible2Vulnerability"/>
</SubObjectPropertyOf>
<SubObjectPropertyOf>
  <ObjectPropertyChain>
    <ObjectProperty IRI="owlapi:ontology578765402008569#foundIn"/>
    <ObjectInverseOf>
      <ObjectProperty IRI="owlapi:ontology578765402008567#includesComponent"/>
```

```
        </ObjectInverseOf>
        <ObjectProperty IRI="owlapi:ontology578765402008577#supportsFunction"/>
      </ObjectPropertyChain>
      <ObjectProperty IRI="owlapi:ontology578765402008571#risksFunction"/>
    </SubObjectPropertyOf>
    <SubObjectPropertyOf>
      <ObjectPropertyChain>
        <ObjectProperty IRI="owlapi:ontology578765402008569#foundIn"/>
        <ObjectInverseOf>
          <ObjectProperty IRI="owlapi:ontology578765402008567#includesComponent"/>
        </ObjectInverseOf>
        <ObjectProperty IRI="owlapi:ontology578765402008581#accessInfo"/>
      </ObjectPropertyChain>
      <ObjectProperty IRI="owlapi:ontology578765402008573#risksInfo"/>
    </SubObjectPropertyOf>
    <SubObjectPropertyOf>
      <ObjectPropertyChain>
        <ObjectProperty IRI="owlapi:ontology578765402008571#risksFunction"/>
        <ObjectInverseOf>
          <ObjectProperty IRI="owlapi:ontology578765402008577#supportsFunction"/>
        </ObjectInverseOf>
      </ObjectPropertyChain>
      <ObjectProperty IRI="owlapi:ontology578765402008588#risksVia"/>
    </SubObjectPropertyOf>
    <SubObjectPropertyOf>
      <ObjectPropertyChain>
        <ObjectProperty IRI="owlapi:ontology578765402008573#risksInfo"/>
        <ObjectInverseOf>
          <ObjectProperty IRI="owlapi:ontology578765402008581#accessInfo"/>
        </ObjectInverseOf>
      </ObjectPropertyChain>
      <ObjectProperty IRI="owlapi:ontology578765402008588#risksVia"/>
    </SubObjectPropertyOf>
  </Ontology>
```

## Appendix B. Inferred Assertions by the Reasoner (OWL Format)

```
<ClassAssertion>
  <Class IRI="owlapi:ontology578765402008551#Risk"/>
  <NamedIndividual IRI="http://www.co-ode.org/ontologies/ont.owl#Log4shell"/>
</ClassAssertion>
<ClassAssertion>
  <Class IRI="owlapi:ontology578765402008553#CybersecurityRisk"/>
  <NamedIndividual IRI="http://www.co-ode.org/ontologies/ont.owl#Log4shell"/>
</ClassAssertion>
<ClassAssertion>
  <Class IRI="owlapi:ontology578765402008555#VulnerabilityInducedRisk"/>
  <NamedIndividual IRI="http://www.co-ode.org/ontologies/ont.owl#Log4shell"/>
</ClassAssertion>
<ObjectPropertyAssertion>
  <ObjectProperty IRI="owlapi:ontology578765402008590#susceptible2Vulnerability"/>
  <NamedIndividual IRI="http://www.co-ode.org/ontologies/ont.owl#App2"/>
  <NamedIndividual IRI="http://www.co-ode.org/ontologies/ont.owl#Log4shell"/>
</ObjectPropertyAssertion>
<ObjectPropertyAssertion>
```

```
                <ObjectProperty IRI="owlapi:ontology578765402008590#susceptible2Vulnerability"/>
                <NamedIndividual IRI="http://www.co-ode.org/ontologies/ont.owl#App3"/>
                <NamedIndividual IRI="http://www.co-ode.org/ontologies/ont.owl#Log4shell"/>
            </ObjectPropertyAssertion>
            <ObjectPropertyAssertion>
                <ObjectProperty IRI="owlapi:ontology578765402008590#susceptible2Vulnerability"/>
                <NamedIndividual IRI="http://www.co-ode.org/ontologies/ont.owl#App4"/>
                <NamedIndividual IRI="http://www.co-ode.org/ontologies/ont.owl#Log4shell"/>
            </ObjectPropertyAssertion>
            <ObjectPropertyAssertion>
                <ObjectProperty IRI="owlapi:ontology578765402008571#risksFunction"/>
                <NamedIndividual IRI="http://www.co-ode.org/ontologies/ont.owl#Log4shell"/>
                <NamedIndividual IRI="http://www.co-ode.org/ontologies/ont.owl#OpenAccount"/>
            </ObjectPropertyAssertion>
            <ObjectPropertyAssertion>
                <ObjectProperty IRI="owlapi:ontology578765402008573#risksInfo"/>
                <NamedIndividual IRI="http://www.co-ode.org/ontologies/ont.owl#Log4shell"/>
                <NamedIndividual IRI="http://www.co-ode.org/ontologies/ont.owl#ClientIDsList"/>
            </ObjectPropertyAssertion>
            <ObjectPropertyAssertion>
                <ObjectProperty IRI="owlapi:ontology578765402008588#risksVia"/>
                <NamedIndividual IRI="http://www.co-ode.org/ontologies/ont.owl#Log4shell"/>
                <NamedIndividual IRI="http://www.co-ode.org/ontologies/ont.owl#App3"/>
            </ObjectPropertyAssertion>
            <ObjectPropertyAssertion>
                <ObjectProperty IRI="owlapi:ontology578765402008588#risksVia"/>
                <NamedIndividual IRI="http://www.co-ode.org/ontologies/ont.owl#Log4shell"/>
                <NamedIndividual IRI="http://www.co-ode.org/ontologies/ont.owl#App4"/>
            </ObjectPropertyAssertion>
        </Ontology>
```

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
