# Peer review of "Sustainable Risk Identification Using Formal Ontologies†"

_algorithms, doi:10.3390/a15090316_

Round 1

Reviewer 1 Report

A very interesting paper. Suggest minor revisions.

Reviewer 2 Report

Method's succesful appliance directly depends on the data sources available and ran and authors have to clearly define how do they mean to collect the data for the solution to work in a proper and reliable manner.

After the case study presented, results shown and the discussiom performed  paper lacks of conclusion part which must be introduced as the separate paragraph at the end of paper. It has to contain strenghts and weaknesses  of the proposed novel solution disclosed and elaborated (according to authors view) and the directions for future research based on the proposed method (what authors aim to do with this method next).
